# New Perspective on Aqueous Humor Circulation: Retina Takes the Lead

**DOI:** 10.3390/ijms26062645

**Published:** 2025-03-14

**Authors:** Satoshi Ueki, Yuji Suzuki

**Affiliations:** 1Center for Integrated Human Brain Science, Brain Research Institute, Niigata University, Niigata 951-8585, Japan; yuji-s@bri.niigata-u.ac.jp; 2Division of Ophthalmology and Visual Science, Graduate School of Medical and Dental Sciences, Niigata University, Niigata 951-8510, Japan; 3Department of Pediatrics, NHO Niigata National Hospital, Kasiwazaki 945-8585, Japan

**Keywords:** retinal glymphatic pathway, aqueous humor, aquaporin-4, aquaporin-1, tight junction, age-related macular degeneration, glaucomatous optic neuropathy, glymphatic flow

## Abstract

Ocular aqueous humor plays an important role in maintaining retinal function. Recent findings indicate that aqueous humor, which flows into the vitreous body, is probably absorbed by Müller cells in the retina, and this process is mediated by aquaporin-4. In this review, we aim to summarize the results of studies on classical aqueous humor circulation and postiridial flow, a pathway proposed in the late 1980s for the inflow of aqueous humor into the vitreous body. In addition, we aim to discuss the retinal glymphatic pathway, inferred by recent findings, with a focus on the anatomical location of aquaporins and barriers that regulate water movement within the tissue. Similarly to the cerebral glymphatic flow, the function of the retinal glymphatic pathway may decline with age, as supported by our findings. In this review, we also discuss age-related ocular diseases that might be associated with the dysfunction of the retinal glymphatic pathway.

## 1. Introduction

Understanding the dynamics of ocular aqueous humor is crucial for maintaining healthy eye function and understanding the pathogenesis of ocular diseases. Most of the ocular aqueous humor produced in the ciliary epithelium (CE) flows into the anterior chamber. It drains out of the eye through the trabecular meshwork (TM) and uveoscleral outflow route. However, some aqueous humor flows into the vitreous body and drains through the retinal pigment epithelium (RPE) into the choroid. This flow, known as the postiridial flow, was proposed by Tsuboi in 1987 [1]. However, the details of this pathway remain unknown. Recently, it was reported that retinal absorption regulates the postiridial flow of aqueous humor into the vitreous body [2]. Furthermore, the absorption was shown to be mainly mediated by aquaporin (AQP)-4 (AQP4), a water channel in retinal Müller cells (MCs) [2]. These findings suggest that the retina is a leading actor in aqueous humor circulation. In the cerebrum, a system called the cerebral glymphatic flow, which is centered on AQP4 in astrocytes, plays a crucial role in flushing out waste products, such as amyloid-β [3,4]. In this review, we aim to describe a new pathway for ocular aqueous humor flow from the vitreous body into the retina, called the retinal glymphatic pathway, based on an analogy of the cerebral glymphatic flow—a topic that has recently gained attention in brain science. Despite the histological and physiological differences between the retina and cerebral cortex, they have many common features, including the absence of a lymphatic system, synapse formation within the tissue to create neural networks, the use of similar neurotransmitters for neuronal communication, and neurons surrounded by glial cells [5]. In addition, we aim to present a classical perspective on aqueous humor circulation, discuss recent studies on aqueous humor inflow into the vitreous body, and investigate the mechanism of this pathway, offering a new perspective on aqueous humor circulation. Furthermore, we aim to discuss ocular diseases that might be associated with dysfunction of the retinal glymphatic pathway.

## 2. A New Perspective on Aqueous Humor Circulation

### 2.1. A Classical Perspective on Aqueous Humor Circulation

Aqueous humor is produced in the CE, located at the posterior part of the iris root, and plays an important role in maintaining ocular homeostasis [6]. Classically, aqueous humor exits the posterior chamber, passes through the pupil into the anterior chamber, then drains through Schlemm’s canal through the TM of the anterior chamber (TM route) or intercellular spaces in the iris root, ciliary muscle fibers, and other uveal structures (uveoscleral route) (Figure 1) [6]. The hydrostatic intraocular pressure is the driving force for drainage [6]. Although the inflow of aqueous humor into the vitreous body is considered to be minimal, the composition of the vitreous body—mostly water with a few percent collagen and hyaluronic acid [7]—suggests that aqueous humor can move freely into it.

Accumulating evidence supports the idea that some aqueous humor flows into the vitreous body (that is, entering of postiridial flow). Araie et al. studied this through animal experiments [8]. Since the inflow rate of the postiridial flow, as calculated from the experimental results, is considerably lower than that into the anterior chamber, it plays a minor role. In contrast, Smith et al. emphasized the importance of this outflow, predicting that the outflow rate at 15 mmHg for a normal human eye is approximately 6 μL/min [9]. Their new theory predicts that the outflow facilities and total pressure-dependent outflow for the whole eye are more than twice as large as the conventional estimate [10]. Therefore, considerations about the postiridial flow remain controversial.

### 2.2. Proposal of the Retinal Glymphatic Pathway Analogy to the Cerebral Glymphatic Flow

AQPs are membrane proteins that selectively allow water molecules to pass through, acting as water channels in living organisms [11]. In mammals, the AQP family consists of 13 members [12]. Since AQPs are important for water dynamics of the posterior eye, and the retina is considered part of the central nervous system, we searched the literature for the distribution of AQP4 and AQP1 in the eye [13,14,15,16,17,18,19,20,21,22,23], which are particularly important for the water dynamics in the central nervous system [24]. We found that AQP4 is specifically distributed in the vitreous side of foot processes in retinal MCs [14], and AQP1 is expressed in photoreceptors and RPEs [17,18]. Furthermore, since AQP4 plays a central role in the cerebral glymphatic flow [24], a new mechanism to explain cerebral water dynamics [3,4], we hypothesized that AQP4 is involved in water dynamics in the posterior eye, similar to its role in the cerebral glymphatic flow, where it facilitates the movement of cerebrospinal fluid across the intraparenchymal intercellular space [3,4]. This process plays a role in flushing out waste products in the intercellular spaces. The relationship between the cerebral glymphatic flow and Alzheimer’s disease, the most common form of dementia, has been widely studied [25,26]. The cerebral glymphatic flow originates from cerebrospinal fluid, which flows into the cerebral cortex through the Virchow–Robin space (VRS) surrounding the peripheral arterial capillaries of perforating branches. Then, it moves into the VRS around venous capillaries and finally drains into the cerebrospinal fluid space [3,4]. Astrocyte AQP4 plays a major role in facilitating cerebral glymphatic flow [24]. Similarly, if water is absorbed in the retina, it would probably play a similar role. We have termed the absorption of the aqueous humor into the retina, mediated by AQP4 function in MCs, the retinal glymphatic pathway, as demonstrated in a previous study [2]. This pathway describes the flow of water with distinct pre- and post-retinal routes (Figure 1).

Findings from a previous study have triggered a reinterpretation of the concept of the postiridial flow. This study used JJ vicinal coupling proton exchange magnetic resonance imaging (JJVCPE MRI) with H_2_^17^O to evaluate water dynamics in the posterior eye of mice [2]. The JJVCPE MRI using H_2_^17^O is a method for evaluating cerebrospinal fluid dynamics in mice [27]. In our previous study, we injected a saline solution containing 40% H_2_^17^O into the femoral vein of a mouse, and the vitreous signal changed within a short time. This evidence suggests that water injected into the vein flows into the vitreous body [2]. Deng et al. reported an inflow of water injected into peripheral vessels into the vitreous body in deuterium oxide (D_2_O) MRI [28]. Based on previous ocular physiological findings, the aqueous humor produced by the CE is the primary source of water flowing into the posterior eye (that is, entering of the postiridial flow), and it originates from peripheral blood vessels [6]. Next, we considered where the aqueous humor flows once it enters the vitreous body. Experimental results indicate that aqueous humor, which flows into the vitreous body, is absorbed by the retina [2]. Furthermore, the authors discovered that changes in the vitreous signal were greater in AQP4 knockout mice compared with those in the control mice, suggesting that AQP4 in the foot processes of MCs plays a crucial role in water absorption into the retina. This finding indicated that the lack of AQP4 in the foot processes of MCs in the retina reduces water absorption into the retina, leading to increased vitreous signal changes [2].

### 2.3. Aquaporins and Barriers Regulate Aqueous Humor and Cerebrospinal Fluid Circulations

Water can move freely through intercellular spaces. However, if intercellular tight junctions (TJs) exist as barriers, water can only move between spaces separated by barriers through the AQPs of the cells. Here, AQPs act as gates.

Cerebrospinal fluid is produced in the choroid plexus and absorbed into the arachnoid villi of the subarachnoid space [4]. However, the discovery of cerebral glymphatic flow has reshaped our understanding of cerebrospinal fluid circulation. AQPs and the brain barrier play essential roles in cerebrospinal fluid circulation, including cerebral glymphatic flow. Of the AQPs, AQP4 and AQP1 play important roles in cerebrospinal fluid circulation [24]. AQP4 is distributed within the brain barrier (collective term for blood–brain barrier, blood–cerebrospinal fluid barrier, and outer brain barrier). These barriers consist of TJs between cerebral capillary endothelial cells or choroid plexus epitheliums or arachnoid barrier cells and the surrounding cells and structures that maintain their function [29]. The role of TJs is significant in terms of their water impermeability [24]. AQP4 is distributed in the glia limitans externa, peri-capillary space of astrocytes, and ependymal cell membranes. Water molecules move freely through intercellular spaces, but TJs restrict the movement of water molecules unless AQPs are present. TJs are composed of many proteins, mainly occludins and claudins [30]. AQP1 is distributed in the choroid plexus epithelium, suggesting its participation in cerebrospinal fluid production [24]. Furthermore, it is distributed in the endothelium of common capillaries but not in the endothelial cells of brain capillaries [24].

The aqueous humor circulation is then discussed in comparison to the cerebrospinal fluid circulation. First, we considered the existence of the eye barrier, which is a structure that does not allow water to pass within the eye, and it functions similarly to the brain barrier. However, it differs from the cerebrum in some respects due to the unique structure of the eye. In this review, barriers are defined as water-impermeable intercellular TJs or tight-like and adherence junctions. Aqueous humor flows into the posterior chamber and vitreous body, and all tissues bordering the anterior chamber, posterior chamber, and vitreous body have barriers [31,32,33,34,35,36,37,38] (Figure 2A). The ocular equivalent of the blood–brain barrier is the inner blood–retinal barrier (IBRB). The IBRB consists of TJs between retinal capillary endothelial cells [32] and the surrounding cells and structures that maintain its function. The blood-aqueous barrier, made up of TJs between the CE cells, which produces aqueous humor, is the ocular equivalent of the blood-cerebrospinal fluid barrier, which produces cerebrospinal fluid. In addition, the outer blood–retinal barrier (OBRB) is similar to the outer brain barrier. OBRB consists of TJs between RPEs [31], and the surrounding cells and structures that maintain its function. TJs between lens epithelial cells, corneal endothelial cells, endothelial cells in the iris capillary, and iris pigment epitheliums are also part of the eye barrier [33,34,36,37]. The role of these barriers is not described in detail, as this review focuses on barriers and AQPs in the retina. In addition to the OBRB and the TJs between RPEs described above, the retina has another barrier, the outer limiting membrane (OLM), which consists of tight-like and adherence junctions between photoreceptors and the foot processes of MCs [31]. Reportedly, TJs between corneal endothelium are incomplete [39], and TJs are present in Schlemm’s canal but with a loose structure [40].

Next, we discuss the anatomical localization of AQPs, which is closely related to the eye barrier. AQP4 and AQP1 are distributed in similar locations [13,14,15,16,17,18,19,20,21,22,23] (Figure 2B). AQP1 is present in corneal endothelial cells, lens epithelial cells, TM, CE, and retina [13,17,18,19,20,21], whereas AQP4 is present in the TM, CE, and retina, overlapping partially with AQP1 distribution [13,14,15,16,22]. CE is responsible for aqueous humor production. A previous study demonstrated that AQP1-null mice experience greater intraocular pressure reduction than AQP4-null mice [41]. Other studies indicated that AQPs have no significant effect on the outflow of aqueous humor through the TM and uveoscleral routes [21], suggesting that AQP1 is more likely to contribute to aqueous humor production. AQP4 is expressed in MCs and astrocytes of the retina. In MCs, AQP4 is unevenly distributed in the vitreous side of the foot process (forming the inner limiting membrane [ILM]) and around retinal capillaries [14], whereas AQP1 is expressed in photoreceptors and the RPE of the retina [17,18]. The roles of AQPs other than AQP4 and AQP1 require further discussion.

## 3. Water Dynamics in the Retina

### 3.1. Three Barriers and Two AQPs

The retina contains three barriers and two AQPs. The three barriers are the IBRB, OBRB, and OLM. The two AQPs are AQP4 and AQP1. These structures regulate water flow within the retina (Figure 3A). The aqueous humor flows into the retina through two pathways involving AQP4 (Figure 3B). The first pathway involves the absorption of aqueous humor from the vitreous body through AQP4 at the foot processes of MCs aligned along the vitreous side [2]. The second pathway involves the movement of aqueous humor into the intercellular space of the retina, which is guided by the first pathway because ILM lacks a barrier. These pathways represent the inflow segment of the retinal glymphatic pathway. Two pathways involving the outflow of water from the retina mediated by AQP4 are proposed for the outflow (Figure 3C). The first pathway involves the outflow of water into the space between the retinal capillary and the foot processes of MCs and astrocytes around the retinal capillary, which is similar to the cerebral glymphatic flow [3,4]. The second pathway guides water from the intraretinal intercellular spaces into the space between the retinal capillary and the foot processes of MCs and astrocytes drawn by the first pathway. Water flow within intercellular spaces plays an important role in removing intraretinal waste products. The intraretinal pathway of aqueous humor through AQP4, expressed on the foot process of the MCs (retinal glia), is called the retinal glymphatic pathway. Since another pathway is required for the outflow of water from the retina, we speculate that a pathway through AQP1 may exist (Figure 3D). In this pathway, water passes through the photoreceptor through AQP1 and drains into the space between RPE cells and photoreceptors, then exits to the choroid through AQP1 in the RPE. Water movement between spaces separated by OLM and OBRB relies on AQP1.

### 3.2. Retinal AQP-4 in the Pericapillary Space

The outflow pathway of water into the space between the foot processes of the MC and astrocytes, as well as the retinal capillary, is discussed here using an analogy of the cerebral glymphatic flow. The cerebral glymphatic flow originates from cerebrospinal fluid, which flows into the cerebral cortex through the VRS around the arterial capillaries at the ends of perforating branches, then flows into the VRS surrounding venous capillaries, and finally drains into the cerebrospinal fluid space [3,4]. According to Katoozi et al. and Ramírez et al., the intraretinal capillaries are surrounded by the foot processes of MCs and astrocytes [15,16]. Using an analogy of the cerebral glymphatic flow, aqueous humor might flow into the retina and subsequently flow into VRS-like structures surrounding the intraretinal capillaries. This raises the question: Where does the flow within these VRS-like structures surrounding the capillaries of the venous system go in the mechanism of the retinal glymphatic pathway? Mathieu et al. demonstrated that dye (fluorescein dextran) injected into the cerebrospinal fluid space enters the optic nerve and moves to vessels and astrocytes within the optic nerve [42]. They proposed the presence of a glymphatic flow within the optic nerve [42], suggesting that water originating from the retinal glymphatic pathway flows into the subarachnoid space around the optic nerve through the space surrounding the central retinal vein, as in the mechanism of the cerebral glymphatic flow.

### 3.3. Driving Force of the Retinal Glymphatic Pathway

As previously discussed, we proposed that two pathways each are responsible for inflow and outflow into the retina. The two inflow pathways are mediated by AQP4 in the foot process of MCs and through the intraretinal intercellular space (Figure 3B). The two outflow pathways involve VRS-like structures through AQP4 in the foot processes of MCs and through the intercellular space (Figure 3C). What is the driving force behind these pathways, the retinal glymphatic pathway? In view of the analogy of cerebral glymphatic flow, Nakada et al. proposed that unidirectional water flow through AQP4 in astrocytes drives cerebral glymphatic flow [24].

The driving force behind this flow produces a retrograde movement in the VRS surrounding the capillaries of the arterial system and an orthograde flow around the capillaries of the venous system. However, the pulsation of the arterial system counteracts the retrograde flow around arterial capillaries, enhancing the flow around the capillaries of the venous system [24]. Cerebrospinal fluid flows into the intercellular spaces of the cerebral cortex as it is drawn by the flow of water into astrocytes from the brain surface. Similarly, water in intercellular spaces flows into the VRS as it is drawn by the flow from the astrocytes into the VRS, which is also true for the retina. This flow is crucial for draining waste materials from the cerebrum. This theory provides a simple and theoretically consistent explanation of the cerebral glymphatic flow. However, the unidirectional flow of water through AQP4 within astrocytes—from foot processes on the brain surface to those on the side of the VRS—has not been proven. The directionality of this flow may be influenced by osmotic pressure, a factor that determines AQP direction [43], or by other unknown factors. Further research and the development of suitable methodologies are needed to validate this hypothesis.

### 3.4. Retinal AQP-1 in Photoreceptors and RPE

AQP1 is expressed in photoreceptors and RPE cells [17,18]. Another pathway for retinal water outflow involves AQP1 in photoreceptors, where water exits into the space between the RPEs and photoreceptors (Figure 3D). From this space, water may flow into the choroid through AQP1 in the RPE (Figure 3D). Given that photoreceptors are constantly responding to light through biochemical and electrophysiological processes, they require constant water absorption and expulsion. This process may serve purposes such as cooling or discharging reactive oxygen species. The direction of water inflow and outflow through AQP1 in the photoreceptors and RPE remains unclear. However, the unidirectional flow of water from the retina to the choroid through AQP1 is a more plausible explanation than the flow of water from the choroid to the photoreceptors through RPEs and then to the retina. This perspective can also explain the concept of the postiridial flow. Since these experiments were performed by injecting high molecular weight dyes such as fluorescein into the vitreous body, these dyes may be trapped by hyaluronic acid and collagen in the vitreous body [8], leading to a low inflow rate. The postiridial flow can be explained by combining the retinal glymphatic pathway through AQP4 in the retinal MCs and the outflow pathway through AQP1 in photoreceptors and RPE. Mosely et al. injected tritiated water (^3^H_2_O) into the vitreous body of rabbits and analyzed its distribution [44]. Blood samples from the vortex vein were collected at 2 min intervals for up to 80 min, whereas anterior chamber fluid was collected at 15 min intervals for up to 240 min after injection [44]. The study reported that most (97%) ^3^H_2_O was detected in the vortex vein, indicating migration from the vitreous body into the choroid. ^3^H_2_O, like H_2_^17^O and [^15^O]H_2_O, is water that is not likely to be trapped in hyaluronic acid or collagen in the vitreous body, like macromolecular dyes. These results support the role of the retinal glymphatic pathway and AQP1-mediated water outflow in photoreceptors and RPE.

## 4. Role of the Retinal Glymphatic Pathway

### 4.1. Age-Dependent Changes in Water Inflow Regulation into the Vitreous Body

A previous study demonstrated that in humans, as in mice, intravenously injected H_2_O flows into the vitreous body [45]. In humans, water dynamics in the posterior eye were evaluated using positron emission tomography with [^15^O]H_2_O instead of JJVCPE MRI with H_2_^17^O [45]. Since no known human diseases occur due to lack of AQP4 (neuromyelitis optic spectrum disorders destroy AQP4 but do not cause their deletion [46]), we hypothesized that water dynamics in the posterior eye could undergo age-related changes with age-related loss of AQP4 function. Next, we discuss the analogy with the cerebral glymphatic flow as a background to this hypothesis. Aging causes reduced drainage of waste materials (especially amyloid-β) in the cerebrum, which contributes to the higher prevalence of Alzheimer’s disease in the older population. Experimental results also suggest a relationship between aging and functional decline of AQP4 [47] and between aging and cerebral glymphatic flow [48]. These findings raise the possibility of age-related changes in the retinal glymphatic pathway. In evaluating the [^15^O]H^2^O inflow into the human vitreous body, no age-related changes were observed in the degree of inflow at the plateau phase. However, the rate of inflow before reaching the plateau differed with age. This phenomenon can be attributed to a decrease in aqueous humor production by the CE and reduced absorption of water in the retina [45].

### 4.2. Age-Related Macular Degeneration and Glaucomatous Optic Neuropathy

The cerebral glymphatic flow is responsible for flushing waste materials from intercellular spaces. The cerebral glymphatic flow serves as a lymphatic system, hence its name (glia + lymphatic) [49]. Based on the analogy of the cerebral glymphatic flow, we hypothesized that the retinal glymphatic pathway plays a role in flushing waste materials from the intraretinal intercellular space.

Dysfunction of the retinal glymphatic pathway might cause ocular diseases by impairing waste drainage. Amyloid-β, a protein produced in the cerebral cortex, is necessary for synapse formation [50] and is a major protein responsible for Alzheimer’s disease, the most common form of dementia [51]. Amyloid-β is also found in the retina and is a component of intraretinal drusen observed in patients with age-related macular degeneration (AMD) [52]. Physiologically, sleep promotes cerebral glymphatic flow [53,54], suggesting that it might enhance the retinal glymphatic pathway. Recent studies indicate a link between sleep disorders and AMD [55]. Various factors may contribute to this association; however, these findings suggest a link between AMD and the retinal glymphatic pathway. Large-scale clinical studies are needed to confirm this relationship; however, a report discovered that 9.5% of patients with AMD and the macular hole or epiretinal membrane (macular hole in 7 eyes and epiretinal membrane in 35 eyes) who underwent vitrectomy, including ILM and epiretinal membrane peeling, experienced exacerbation of choroidal neovascularization [56]. Considering that the foot processes of MCs constitute the ILM, these findings suggest an association between retinal glymphatic pathway damage and AMD progression. In animal models of elevated intraocular pressure, amyloid-β has been detected in the retina, optic nerve, and lateral geniculate body, suggesting an association between glaucomatous optic neuropathy (GON) and amyloid-β [57,58]. The total protein levels in the anterior chamber are higher in patients with glaucoma compared with their levels in the controls, and its concentration correlates with amyloid-β42 level [59]. AMD and GON, like Alzheimer’s disease, become more prevalent with age. Moreover, many studies have shown that glaucoma is more prevalent in patients with Alzheimer’s disease [60]. The evidence showing a link between the retinal glymphatic pathway and AMD or GON is summarized in Table 1. Since dysfunction in the cerebral glymphatic flow is involved in the pathogenesis of Alzheimer’s disease, dysfunction in the retinal glymphatic pathway is involved in AMD and GON pathogenesis. Daruich et al. found that AQP4 is densely expressed in the retina, especially in the central foveal region [31]. The uneven distribution of AQP4 in the XY-plane of the retina may be related to the pathogenesis of ocular diseases, such as AMD and GON, necessitating further research. Further studies will clarify the clinical implications of the retinal glymphatic pathway for patients with these ocular diseases.

## 5. Conclusions

In this study, we reviewed the classical perspective on aqueous humor circulation in the eye and discussed findings related to its inflow into the vitreous body. Furthermore, the possibility of aqueous humor inflow into the retina was discussed based on experimental results, and the concept of the retinal glymphatic pathway, which includes the pathways before and after water inflow into the retina, was proposed based on the analogy with the cerebral glymphatic flow. In addition, the flow of water through the photoreceptors and RPE was discussed. These concepts were derived from detailed considerations of the anatomical relationship between ocular barriers and AQPs. The retinal glymphatic pathway proposed in this study supports Smith et al.’s estimate that outflow rate is approximately twice as high as previously thought [10]. However, further research is needed to determine the actual estimate that considers the routes of the retinal glymphatic pathway. Our hypothesis on aqueous humor circulation in the retina, including the retinal glymphatic pathway, offers a new perspective on aqueous humor dynamics in the posterior eye. The role of this pathway in ocular diseases, such as AMD and GON—both associated with aging and Alzheimer’s disease—is becoming clear, offering new treatment methods. Wang et al. conducted experiments in which amyloid-β was injected into the vitreous body of mice and rats, demonstrating its accumulation in the perivascular space along the optic nerve vein, thereby proposing the ocular glymphatic system [61,62]. Mouse experiments using JJVCPE MRI with H_2_^17^O and human experiments using positron emission tomography with [^15^O]H_2_O showed the same water dynamics as those observed by Wang et al. The ocular glymphatic system described by Wang et al. also shares many similarities with our proposed retinal glymphatic pathway, including AQP4 inhibition and age-related decline in function. Our new hypothesis underscores the critical role of aqueous humor dynamics in posterior eye waste removal and provides a more detailed understanding of this process. As discussed, dysfunction of the retinal glymphatic pathway might be involved in the pathogenesis of ocular diseases such as AMD and GON.

## Figures and Tables

**Figure 1 ijms-26-02645-f001:**
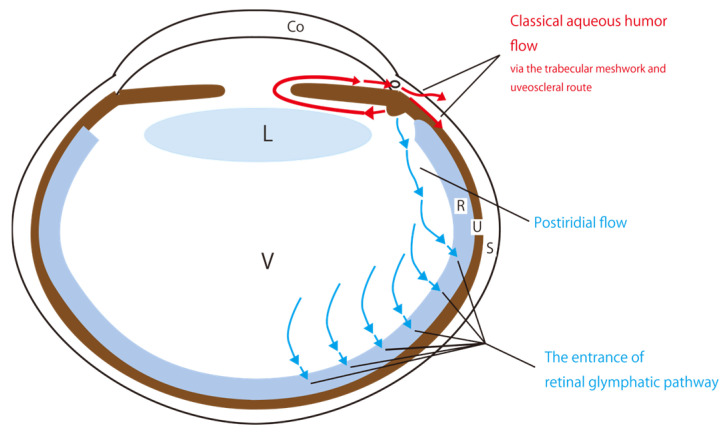
A schematic showing classical and new perspectives on aqueous humor circulation. Classically, aqueous humor exits the posterior chamber and passes through the pupil into the anterior chamber. Subsequently, aqueous humor flows into the Schlemm’s canal through the trabecular meshwork, or intercellular spaces in the iris root, ciliary muscle fibers, and other uveal structures (uveoscleral route) (red arrows). Tsuboi proposed the concept of the postiridial flow in 1987 [1]. We propose a new perspective on aqueous humor circulation, the retinal glymphatic pathway (blue arrows). The inflow of the aqueous humor through the retinal glymphatic pathway is associated with aquaporin-4 function in the Müller cells in the retina. Abbreviations: Co, cornea; L, lens; R, retina; S, sclera; U, uvea; V, vitreous body.

**Figure 2 ijms-26-02645-f002:**
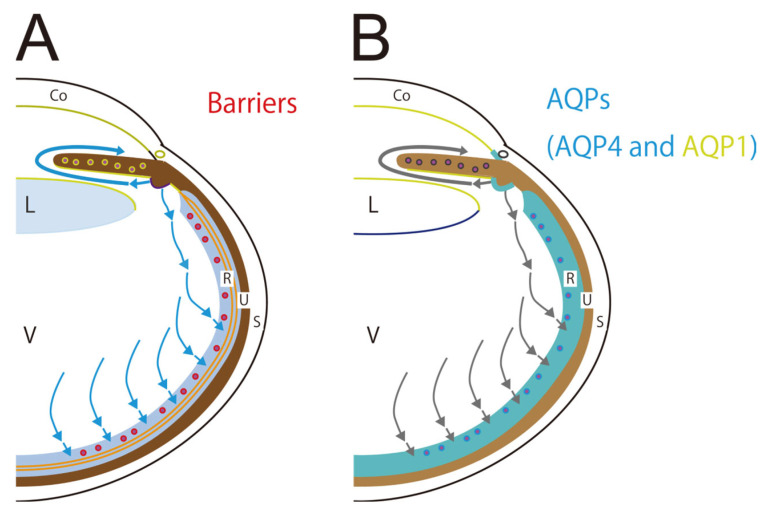
A schematic diagram showing the distribution of barriers, that is, TJs and tight-like and adherence junctions, in the eye (red, orange, purple, and green lines) (**A**). All tissues bordering the anterior chamber, posterior chamber, and vitreous body have barriers attributed to TJs or tight-like and adherence junctions. In the retina, TJs are observed between endothelial cells in the retinal capillary (red lines), between the RPE, and the tight-like and adherence junctions between the photoreceptor and the foot process of Muller cells (orange lines). Blue arrows indicate aqueous humor flows. A schematic showing the distribution of AQP4 and AQP1 in the eye (AQP4, blue; AQP1, green; AQP4 and AQP1, mixed blue and green) (**B**). The anatomical localization of the two AQPs is closely related to the eye barrier: TJs and tight-like and adherence junctions, and AQP4 and AQP1 are distributed in almost the same location. AQP1 is present in the corneal endothelial cells, lens epithelial cells, TM, CE, and retina. AQP4 is present in the TM, CE, and retina. Gray arrows indicate aqueous humor flows. Abbreviations: AQP, aquaporin; AQP1, aquaporin-1; AQP4, aquaporin-4; CE, ciliary epithelium; Co, cornea; L, lens; R, retina; RPE, retinal pigment epithelium; S, sclera; TJ, tight junction; TM, trabecular meshwork; U, uvea; V, vitreous body.

**Figure 3 ijms-26-02645-f003:**
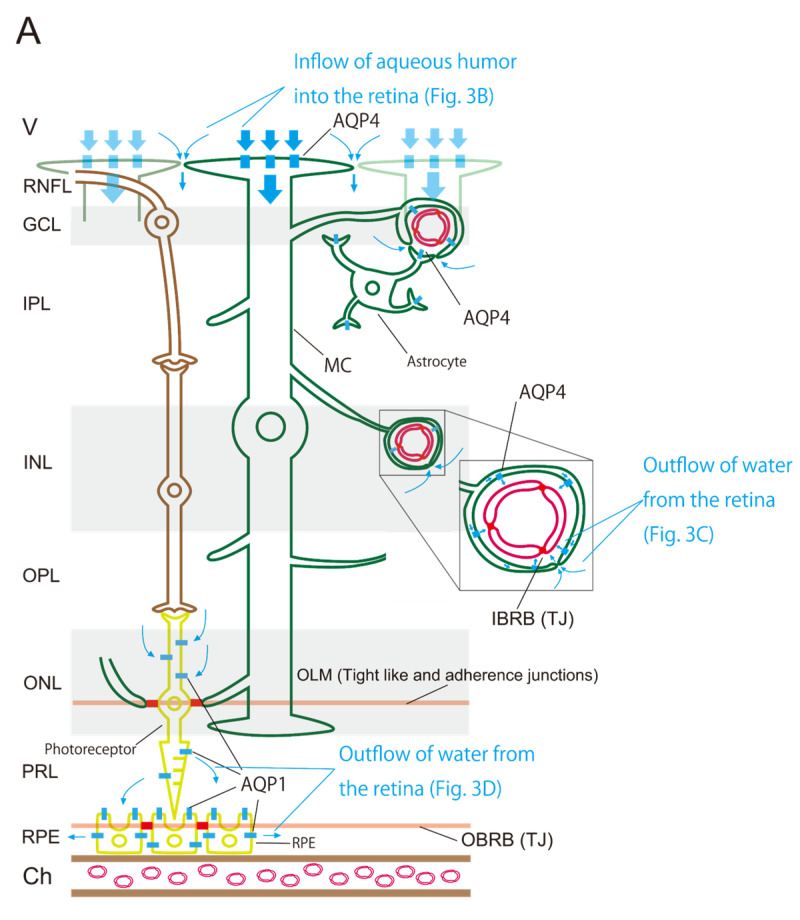
Schematic diagrams showing the distribution of three barriers, two AQPs, and estimated water flows in the retina (**A**). The retina has three barriers, which are the IBRB, OBRB, and OLM (red and orange lines). AQP4 and AQP1 (blue line) are unevenly distributed in the retina. These three barriers and two AQPs regulate water flows in the retina. The inflow of aqueous humor into the retina (**B**). It consists of two pathways: The pathway by which AQP4 at the foot process of the MC aligned on the vitreous side absorbs the aqueous humor that has entered the vitreous body and a pathway in which aqueous humor flows into the intercellular space in the retina guided by the former pathway. The outflow of water from the retina through AQP4 (**C**). It consists of two pathways: One is the outflow through AQP4 into the space between the retinal capillary and the foot processes of MCs and astrocytes around the retinal capillary. Water within retinal intercellular spaces is drawn by the former pathway and flows into the space between the retinal capillary and the foot processes of MCs and astrocytes. The outflow of water from the retina through AQP1 (**D**). The pathway in which water passing through the photoreceptor through AQP1 drains into the space between the RPE and photoreceptor, and water entering this space is expected to flow out to the choroid through AQP1 of the RPE. Blue arrows indicate water flows. Abbreviations: Ch, choroid; GCL, ganglion cell layer; INL, inner nuclear layer; IPL, inner plexiform layer; MC, Müller cell; ONL, outer nuclear layer; OPL, outer plexiform layer; PRL, photoreceptor layer; RNFL, retinal nerve fiber layer.

**Table 1 ijms-26-02645-t001:** The evidence showing a link between the retinal glymphatic pathway and AMD or GON.

AMD	Amyloid-β is a component of intraretinal drusen [52].
	Sleep promotes cerebral glymphatic flow [53,54].
	A link between AMD and sleep disorders [55].
	Inner limiting membrane peeling exacerbates choroidal neovascularization [56].
GON	In intraocular pressure elevation models, amyloid-β is detected in the retina, optic nerve, and lateral geniculate body [57,58].
	The total protein levels in the anterior chamber are higher in patients with GON [59].
	GON is more prevalent in patients with Alzheimer’s disease [60].

AMD, age-related macular degeneration; GON, glaucomatous optic neuropathy.

## Data Availability

No new data were created or analyzed in this study. Data sharing is not applicable to this article.

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
