# Peer review of "New Perspective on Aqueous Humor Circulation: Retina Takes the Lead"

_ijms, 2025, doi:10.3390/ijms26062645_

Round 1

Reviewer 1 Report

Comments and Suggestions for Authors

Ueki & Suzuki review old and new mechanism of vitreous humor production and drainage in the eye. They discuss proposed AQP channel mechanisms for retinal drainage, and how these mechanisms are significant in pathologies such as Alzheimer’s Disease, Sleep disorders, Age-related macular degeneration, and glaucomatous optic neuropathy. The thesis of the paper is sound and so is the support behind it, but extensive revision is required to address important issues. Specifically, there are several instances where information is unnecessarily repeated; this is very confusing for the reader. Additionally, some sentences make incomplete arguments, and citations are oftentimes missing. Finally, the figures need corrections. Specific comments are mentioned by line below. 

MAJOR CONCERNS

  • Line 33 “the retina is the outpost agency of the brain”. What does this mean? Also “the retina and brain differ, but they share many similarities”. What kind of similarities? This is way too general and not supported by any references. I would propose to remove the sentence altogether.
  • 2.1 is very confusing and should be re-written; especially lines 51-64. In line 53 the authors say that the postiridial flow is minimal, and then go on to describe experiments estimating the flow to be up to 6ml/min (a huge volume). So, which one is true?
  • Is the postiridial flow a second way of aqueous humor circulation or is it part of the glymphatic system? As currently written, it appears that it is part of the classical pathway, as it is described in section 2.1. How many classical and updated pathways are there for aqueous humor circulation? This should be clearly stated in text and figures, and the postiridial flow should be described in a separate paragraph or in 2.2.
  • Figure 1 should be updated accordingly. Does the postiridial flow feed into the glymphatic system? This should be made clear in the figure. Matching colors between flows and their respective arrows could help the reader understand.
  • Line 81 “…the vitreous body”, line 87 “…by the retina”. References are missing.
  • Line 93-107. This general description of the previously established brain glymphatic system should appear prior to the introduction of the newer concept of the retinal glymphatic system (lines 75-92)
  • In 2.3 the authors provide an extensive description of AQP family members. However, the role of AQP4 as part of the retinal glymphatic system has been already mentioned in 2.2, which is confusing. A general introduction to AQPs should come prior to the specific roles they play in the retinal glymphatic system, since it is foundational to this system.
  • Line 117, 138; What is the outer brain barrier? Line 127; what is the eye barrier?
  • Line 117. The BBB does not consist of tight junctions. It consists of tight junctions in endothelial cells, plus pericytes, basement membrane and astrocyte endfeet. Please add this important information.
  • Line 133. The same comment is true for the iBRB, which consists of tight junctions in endothelial cells, plus pericytes, basement membrane, and astrocytes/muller glia; and line 137, the oBRB, which also includes the Bruchs membrane and the choroid. Please make sure to describe accurately all the barriers you mention in the manuscript and provide references.
  • Line 135. It is not clear to me why the authors try to make an analogy between the blood-CSF-barrier and the blood-aqueous barrier. What are the similarities between those structures?
  • Figure 2A. The various barriers should be described in detail in different colors. Currently this schematic is not informative at all.
  • Line 143-145. What is the relevance of the information regarding claudin-5? Cld5 is endothelial specific; is it equally expressed on the choroid plexus of the retina and brain, and the rest of CNS blood vessels?
  • Line 146-162. This paragraph is very confusing. The authors attempt to describe AQP1 and 4 expression in the retina. First, it would be useful to update figure 2B, so that each APQ has a different color, and the reader can appreciate different distribution across anatomical locations. Second, the authors claim that the distribution of AQPs in the retina is different than the brain (line 160-161). In what aspect? Shouldn’t they tell us how the distribution looks like in the brain, to appreciate any comparisons?
  • 3.1. I have the same comments as above (comment 9/10). Please be more accurate in your barrier description. This section also contains lots of repeated information from section 2. Line 185: Citation required for claims made here.
  • Please provide reference for the AQP1-mediated outflow pathway (line 199).
  • Figure 3. It would be useful to draw each cell type participating in the fluid flow in the retina in different colors.
  • Figure 3. There are no blood vessels in the OPL of the retina. Please correct the figure to place them in the GCL and INL only.
  • Line 303-307. Please provide reference for this information.
  • Line 309-320. This paragraph is redundant, as its content has been discussed multiple times earlier. In addition, it would fit better earlier in the manuscript.

MINOR CONCERNS

  • The title of the manuscript would be shorter, for example “New perspectives on aqueous humor circulation; retina takes the lead’
  • The authors use past tense throughout the manuscript (for example “we aimed, we discussed” lines 12, 15 ,18, 36, 39, 42 etc.). Please change with present tense.
  • Line 18 “recent findings” please omit or replace, as it is repetitive (see line 15).
  • Line 54. A paragraph break could be used in this location to separate ideas.
  • Figure 1. Please correct Schlem to Schlemm’s canal
  • Line 79 “in our experiment” unusual language tone choice, perhaps consider different words.
  • Line 90; move the reference after line 92 to cite all conclusions mentioned. 
  • Line 109 “CSF is traditionally produced in the choroid plexus…”. Is there a non-traditional production site? Please explain.
  • Line 146; wrong tense for this sentence.
  • Line 239 “in a coexisting manner”. What does this mean? In parallel to the classical pathway? Please rephrase.
  • Line 272-274. Please rephrase this sentence as currently it is difficult to read.
Comments on the Quality of English Language

A language editing is recommended.

Reviewer 2 Report

Comments and Suggestions for Authors

1-      Provide a more details for the dynamics of ocular aqueous humor in the introduction, I would like the authors to clarify this point,

2-      Their new theory predicts 62 that outflow facilities and total pressure-dependent outflow for the whole eye are more 63 than twice as large as the conventional estimate, please clarify the theory?

3-      What is the rationale behind choosing the retinal glymphatic pathway-Analogy to cerebral glymphatic flow? Give much details about What is the significance of studying this pathway.

4-      A more detailed explanation is required regarding AQPs ?

5-      A table summarizing the evidences that linked between the Age-related macular degeneration and glaucomatous optic neuropathy is required.

6-      What are the clinical implications of this study for patients with AMD ?

7-      Manuscript needs language revision and correction of the grammatical and syntax errors.

8-      Where is the discussion and imitations for this study?

Comments on the Quality of English Language

  Manuscript needs language revision and correction of the grammatical and syntax errors.

Round 2

Reviewer 1 Report

Comments and Suggestions for Authors The authors have incorporated all of our comments and any replacement text seems faultless for the issues they address. They also used a professional english editing service and it shows. They made fixes we didn't even recommend so I would say that is clear as well. Overall I have nothing further to add.

Reviewer 2 Report

Comments and Suggestions for Authors

Good job